# Kittel's Brief Comments Endorsing an Alternative View on Barium Titanate

**Akira Kojima**

Fundamental Research Institute for Ferroelectrics (FRIFF), 1439 Yabu, Yasu, Shiga 520-2433, Japan;
akrkojima@ybb.ne.jp

**Abstract:** Charles Kittel has written a masterpiece book, "Introduction to Solid State Physics" (ISSP). He mentions in the chapter on ferroelectrics in detail that barium titanate is the typical displacive-type ferroelectric compound where the $Ti^{4+}$ displacement develops a dipole moment, which has made a deep impression in our mind. The author's group, however, has arrived at an alternative viewpoint on the unit cell structure of barium titanate based on their exhaustive experimental studies. Accordingly, the author sent his relevant papers in 2006 and 2007 to Kittel. He endorsed the results frankly with reminiscence. He mentioned revising the ferroelectric chapter of ISSP according the author's suggestions. It appears to be admissible to publish details now after Kittel has passed away. A long time misunderstanding of the phase transition in barium titanate is due to the text book knowledge of ISSP.

**Keywords:** Charles Kittel; brief comments; barium titanate; alternative view; endorsement





## 1. Introduction

We have lost a great scientist in 2019. Charles Kittel died on 15 May at the age of 102. He is very well known, not only as an eminent theoretical physicist who presented the RKKY interaction [1] and the Kittel magnon mode in ferromagnets [2], but also as a talented author of the well-known books: "Introduction to Solid State Physics" (ISSP), "Quantum Theory of Solids", and "Thermal Physics" (TP). ISSP and TP are unique and excellent because Kittel has looked at solid state in the reciprocal space as much as possible. ISSP was edited eight times from the 1st edition in 1953 to the 8th edition in 2005. He revised the contents and added new chapters by frequent editing, but as far as the ferroelectric chapter is concerned, he made few revisions, mainly due to little noticeable progress of the experimental research in this field.

Concerning the description of barium titanate $BaTiO_3$, he explains the $Ti^{4+}$ displacement in the unit cell of the ferroelectrics phase, and mentions that this displacement develops a dipole moment. However, he does not reveal the structure figure in which the concrete distance of the $Ti^{4+}$ displacement is expressed based on some reliable experimental measurement, but merely shows an illustration for the purpose of its explanation [3] (p. 470). In fact, looking at the figure caption attentively, we can notice it has no mention of the researchers' name, different from many other figures. Accordingly, for more than seventy years, many readers may have been confused into thinking of the figure as if it were real data [4] in the case where they do not pay special attention.

In principle, the exact unit cell information can be obtained from the X-ray diffraction study. Examining anew previous experimental results of the structure by X-ray diffraction, the structure formally reported appears to be incorrect. More than seventy years ago, Megaw studied a powder sample of barium titanate, using a diffractometer [5]. She concluded that barium titanate has a tetragonal unit cell at room temperature. The method she used is useful in many cases, but for barium titanate it is not suitable, the reason for which will be explained later. B.T. Matthias and A. von Hippel also performed X-ray diffraction measurements using a precession camera [6]. Although they observed an

essential anomaly, taking a precession photo, they did not arrive at the correct structure, but simply concluded to have a tetragonal unit cell having twin structure, following Megaw's results. Since then, plenty of researchers have made efforts to study barium titanate by various methods except the X-ray diffraction measurements [7], but strictly speaking, they could not clarify the essential structural nature of barium titanate.

## 2. Notable X-ray Diffraction Results and Kittel's Comments on Them

In order to clarify the essential properties of barium titanate, the author's group has used two different experimental studies in parallel. One is a direct and precise in situ observation of the phase transition process from the paraelectric phase to the ferroelectric one, developing the MKS-cell (Milli-Kelvin stabilized cell) [8–10]. The other is the precise X-ray diffraction study using a precession camera. Since in the early stage of the study using the MKS-cell, single crystals were not chemically etched by phosphoric acid to eliminate some distortion of the surface caused by cutting, but used as-cut state for the measurements, it was easily found that barium titanate has two thermal anomalies at the phase transition on cooling: The temperature of the second thermal anomaly appeared 0.25 K apart from that of the first one [10]. As will be mentioned later, the two thermal anomalies are observed in a very small temperature range as long as 1 mK in an etched single crystal having no restriction on the surface. In accordance with this observation, two kinds of Bragg spots were observed in the X-ray precession photographs, one of which is shown in Figure 1 [11].

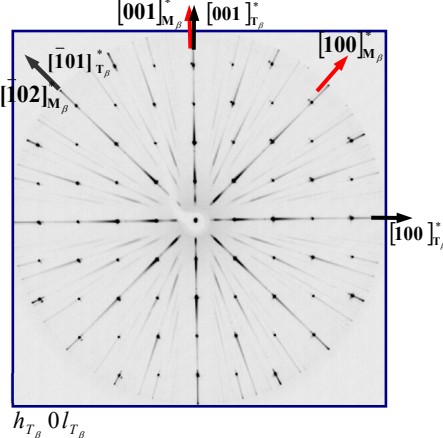

**Figure 1.** X-ray precession photograph of barium titanate single crystal taken at room temperature. The extra Bragg spots can be recognized very close to the main Bragg spots in the first and the third quadrants [11]. β denotes the room temperature phase. Streak lines are due to the continuous X-ray, so that they disappear if an appropriate filter is used.

We can recognize thick main Bragg spots and weak extra Bragg spots near the main spots in the photo. It is well known that one Bragg spot in the reciprocal space corresponds to one plane of a crystal in real space. It is noticed here that all the Bragg spots near the origin of the reciprocal space can be detected by photos of the precession camera without distortion, but in general a diffractometer, which Megaw also used, detects few of the Bragg spots, so that it is impossible for the diffractometer in principle to obtain the reciprocal information completely and exactly without any ambiguity. The intensity of the extra Bragg spot observed is approximately one tenth of the main Bragg spot very close to it. In addition, although the main spots are on a rectangular net, the extra spots are on an oblique net as schematically shown in Figure 2.

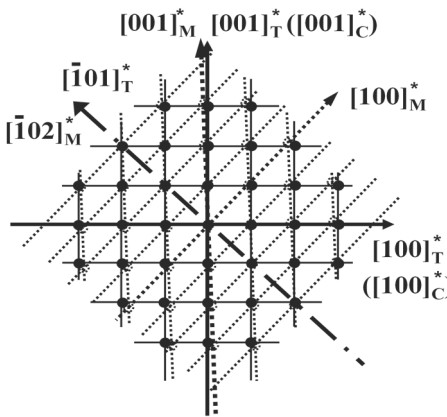

**Figure 2.** Schematic presentation of the precession photo [11]. The main Bragg spots denoted by full circles are on the rectangular tetragonal nets. Open circles imply the extra Bragg spots, and they are on an oblique monoclinic net shown by dotted lines.

These are essentially the same results as the precession data obtained by B. T. Matthias and A. von Hippel [6]. The other two perpendicular reciprocal planes, however, have only one kind of the Bragg spots being on the rectangular line. Using the principle of diffraction, it is possible to obtain the unit cell in real space exactly. The conclusion is barium titanate has a "coherent hybrid structure" as shown in Figure 3.

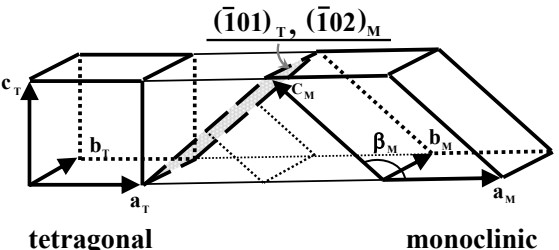

**Figure 3.** Illustration of the obtained "coherent hybrid structure" composed of tetragonal and monoclinic unit cells in the room temperature ferroelectric phase [11]. Two unit cells have a common interface shadowed in the figure, so that they have a structural coherency. Both unit cells have a center of symmetry. $Ba^{2+}$, $Ti^{4+}$, and $O^{2-}$ are on the corner, on the body center, and on the face center of each unit cell, respectively. Their bottom planes are not piled up perfectly, but that of the monoclinic is slightly inclined.

It has two kinds of unit cells composed of a tetragonal and a monoclinic lattice, having a common interface, so that the unit cell has a structural coherency [11]. When the author was giving a lecture of "Crystal structure analysis by X-ray diffraction" to the third year students in his university, he gave them an exercise to obtain the unit cell of barium titanate, giving the precession photo of Figure 1 and those of two perpendicular planes, and most students could correctly obtain the coherent hybrid structure which is essentially the same as Figure 3. It may be the first finding that a solid state compound consists of two structures in one phase. Here, it is insisted that $Ti^{4+}$ is at the position of the body center of both unit cells, so that it can be said there is no $Ti^{4+}$ displacement in each unit cell. The author's group has published a relevant paper in 2006 [11]. The author immediately sent a preprint of the paper to Kittel by e-mail when it was accepted. One week later, he sent his comment to the author by e-mail as follows.

*"Dear Professor Kojima,*

*I have now received and read with much interest your paper with your coworkers on the coherent hybrid structure of BaTiO/3 below the paraelectric transition. For nearly 60 years I have been interested in this phase transition, starting with the work of Wul and then Matthias. Your results are convincing; I only wish there were a simple theoretical model of the transition. In fact it must*

*be one of the most complex transitions known to us. Thank you for your patience in seeing that I received a copy of the paper.*
*Sincerely yours,*
*Charles Kittel"*

The author's group has studied successively the unit cell in all phases, and published the next paper concerning the unit cell evolution of barium titanate in 2007 [12]. Barium titanate has the following unit cells in its four phases: A cubic, a coherent hybrid structure composed of a tetragonal and a monoclinic one, another coherent hybrid of a tetragonal and a monoclinic one, and a tetragonal from high to low temperature as shown in Figure 4.

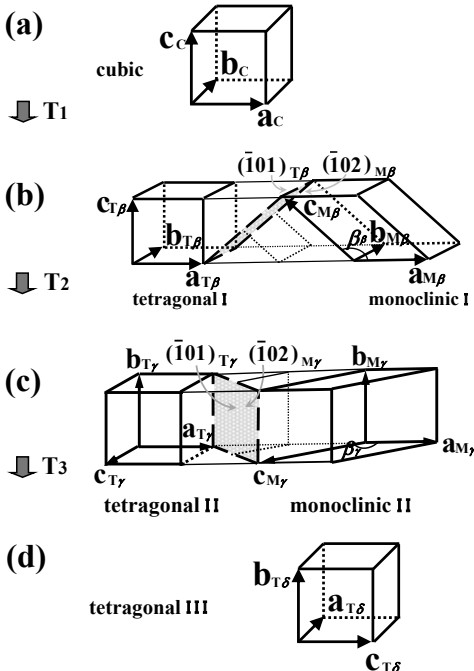

**Figure 4.** View of the unit cell evolution by the three phase transitions obtained by the author's group [12]: (**a**) Cubic lattice in the highest temperature α-phase. (**b**) A coherent hybrid structure composed of a tetragonal $T_I$ and a monoclinic $M_I$ lattices in the room temperature β-phase. The shadowed plane presents a coherent interface. (**c**) Another coherent hybrid structure composed of a tetragonal $T_{II}$ and a monoclinic $M_{II}$ lattices in the lower temperature γ-phase. (**d**) Tetragonal $T_{III}$ lattice in the lowest temperature δ-phase [12].

It can be said that barium titanate does not take the unit cell evolution as the researchers simply think in their heads, i.e., a crystal does not always take lower symmetric structure when its temperature is lowered. It chooses another unit cell evolution mentioned above in order to decrease its Free Energy. It is to be insisted here that such unit cell evolution is not restricted to barium titanate only. Potassium niobate $KNbO_3$ also has the same unit cell evolution as barium niobate [12].

The author sent a preprint of the paper to Kittel by e-mail. Kittel sent the author his comment to the paper by e-mail as follows.
*"Dear Professor Kojima,*
*Many thanks for sending me the reprint of your exhaustive and definitive study of the evolution of the unit cell structures of barium titanate. I remember that some sixty years ago I spent an evening with Bernd Matthias and von Hippel in Cambridge MA when they discussed their new x-ray results. How things have evolved!*
*With kind wishes,*
*Charles Kittel"*

We can easily understand from his book ISSP that Kittel has deeply understood the principle of diffraction, so that he discussed it with B. T. Matthias and A von Hippel in

Cambridge, MA about their results, as mentioned in Kittel's comment. It is guessed that the main point of the discussion was the anomalous extra Bragg spots being on a slightly oblique net. Presumably, from that time at the bottom of his heart, Kittel has been having a doubt about their conclusion that barium titanate has a tetragonal unit cell with a twin structure. Accordingly, Kittel immediately sent the author his comment saying the results obtained by the author's group are convincing.

## 3. Alternative View on Yielding Ferroelectricity at the Phase Transition on Cooing and the Domain Structure in the Ferroelectric Phase

The results of the precise in situ observation of the phase transition from paraelectric phase on cooling and the micro Raman mapping in addition to the X-ray diffraction results have enabled us to arrive at a comprehensive understanding of the evolution of ferroelectricity in barium titanate. Figure 5 shows the results of the measurement of the structural phase transition which occurs on cooling in a very small temperature range within 1 mK [13].

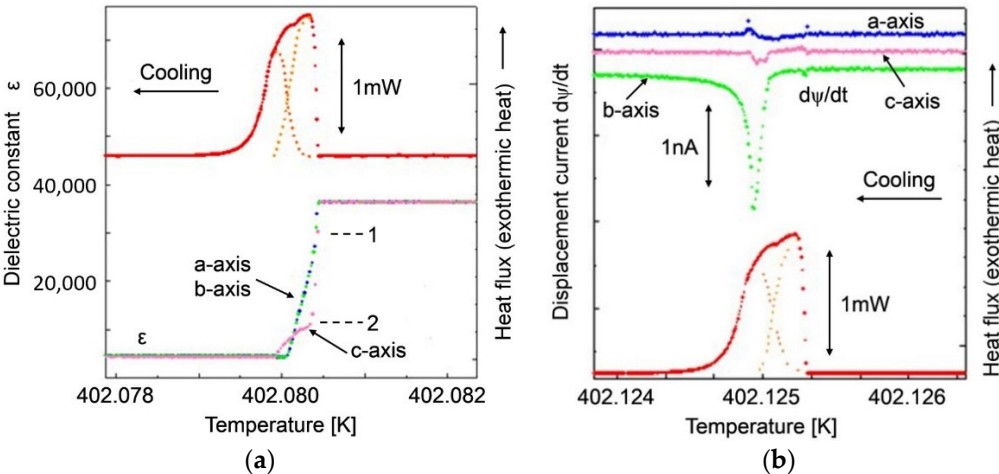

**Figure 5.** (**a**) Simultaneous measurements by the MKS-cell [8–10] of the thermal anomaly with the dielectric constants along the three directions of the crystallographic axis during the phase transition on cooling. The cooling rate is 3 μK/s. (**b**) Simultaneous measurements by the MKS-cell of the heat flux with the displacement current along the three directions of the crystallographic axis during the ferroelectric transition on cooling [13].

At the first anomaly, the dielectric constant reveals the tetragonal nature, but its character disappears at the second thermal anomaly. At the second thermal anomaly, a large displacement current flows along one crystallographic direction, which gives rise to some polarization such as a pyro-charge on both sides of a cubed sample. It should be emphasized that this has been recognized for the first time by the direct measurements of the phase transition on cooling. Barium titanate has surely become ferroelectric. Measurement of the micro Raman mapping reveals that barium titanate has thick and thin domains aligned in parallel in the room temperature phase as shown in Figure 6. The width rate of the thick domains vs. the thin ones is approximately 10:1 [13].

These results are consistent with the X-ray diffraction results, and all of them enrich our understanding of barium titanate. The intensity ratio of the two kinds of Bragg spots is the same as that of the width ratio of the two kinds of domains, so that it is apparent that the coherent hybrid structure recognized by X-ray diffraction is taking the form of thick tetragonal and thin monoclinic domains aligning in parallel. It is thought that barium titanate changes its structure from a cubic to a tetragonal at the phase transition temperature, but the tetragonal structure is unstable due to the comparatively large axial ratio, so that it immediately yields monoclinic domains interstitially inside of it, which causes the displacement current flow. Since it has already been reported by Merz that domains of barium titanate are sensitive to an external electric field [14], it is possible to

imagine that the so-called hysteresis curve is obtained by the increase of the width of the tetragonal domains and the decrease of the monoclinic domains or vice versa in order to reduce the Free Energy under a strong ac electric field. By the way, the author sent a preprint of the latest paper [13] to Kittel by e-mail, but received no answer from him. Since the author knows Kittel's address in Berkeley, he sent the paper to Kittel by air mail with the author's e-mail address added. One month later the author received an e-mail from his son Peter Kittel saying that it was impossible for his father to think about physics anymore because of his age.

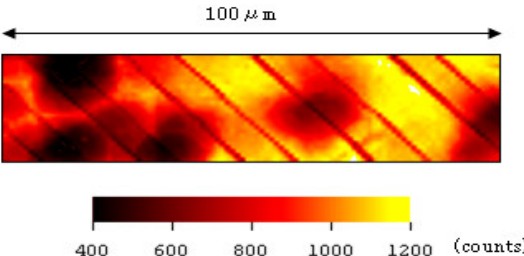

**Figure 6.** Intensity distribution of the preliminary micro Raman mapping near 310 cm$^{-1}$ of the surface of $4 \times 4$ mm$^2$ plate of a barium titanate single crystal at the room temperature. Narrow parallel lines are clearly recognized to have approximately 1 μm width. They are approximately 10 μm apart on average with each other. Dark parts are partly observed; the sample plate is not sufficiently thick, so that it has the possibility to detect some imperfections of the opposite surface of the plate [13].

At present it is impossible to explain theoretically the reason of the flow of the displacement current due to the formation of the monoclinic domains inside the tetragonal domain. The author, however, considers the possibility that it may be due to the imperfection of arrangements of composed atoms at the boundary of the two kinds of domains. In any case, if it could be explained theoretically, the question why ferroelectricity occurs could be clarified, which will grant Kittel his wish saying there were a simple theoretical model of the transition. Even if it's impossible to obtain such theoretical model, Kittel would revise the ferroelectric chapter, using notable experimental results obtained by the author's group, because when the author asked Kittel to send a preprint of the second paper mentioned above [12] in 30 June 2007, Kittel returned his e-mail to the author on the next day as follows.
*"Dear Dr. Kojima,*
*I would like to read your paper. If you would kindly send it to me by post mail as printed matter to ⋯ (his address abbreviated here). I hope at some future date to work on a revision of the ferroelectric chapter of ISSP*
*Sincerely,*
*Charles Kittel"*

### 4. Summary

It has been clarified by his messages to the author by e-mail that Kittel endorsed the alternative unit cell of barium titanate obtained by the author's group and intended to revise the ferroelectric chapter of ISSP in the future.

John Wiley and Sons, the publishing company of ISSP, frankly accepted the author's notification of Kittel's comments. The estimation of ISSP is invariable even if a few mistakes exist, so that its publishing will be continued. Knowledge by ISSP, however, has been deeply impressed on the researchers' mind. Accordingly, it will be expected to devise appending some note of correction in the relevant part of ISSP as soon as possible, for the sake of proper progress of the ferroelectrics study by a new look at ferroelectrics.

**Funding:** This research received no external funding.

**Data Availability Statement:** The data presented in this study are available on request from the corresponding author.

**Acknowledgments:** The author deeply appreciates Annette Bussmann-Holder of the Max Planck Institute for supporting him since the 1990s, evaluating the works by his group.

**Conflicts of Interest:** The author declares no conflict of interest.

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
