# Peer review of "Kittel’s Brief Comments Endorsing an Alternative View on Barium Titanate"

_condensedmatter, doi:10.3390/condmat6010010_

Round 1

Reviewer 1 Report

Author described the phase transition between paraelectric and ferroelectric in BaTiO3. Many papers which are cited in this manuscript have already discuss the phase transition intensively and permitted to be published in their Journal. However, this manuscript just mentioned the results and did not add new things except referring the communication with past professor Kittel. What is the purpose of this manuscript?

Author Response

In the revised manuscript, the author has described the phase transition from paraelectric to ferroelectric in BaTiO3 using three figures cited in the other journal FERROELECTRICS. The author believes that the results of the paper in FERROELECTRICS are superior compared to many papers reported previously, because the measurements were made precisely in a very small rate of temperature change on cooling. The obtained results are consistent with the alternative coherent hybrid structure Kittel has endorsed. Accordingly in this manuscript, new things are added above the communications with Kittel.

Reviewer 2 Report

The author has done a very detailed experimental study and put forward another point of view on the unit cell structure of Kittel's barium titanate. At the same time, the author sent the paper to Kittel via e-mail. Unfortunately, Kittel was unable to reply because of his age, but the publishing company of "ISSP" frankly accepted the author's notification about Kittel's comments. It is expected that the relevant parts will be revised. It can be seen that the author has done a fairly complete description and verification of this paper. I suggest that this paper can be accepted for publication without modification.

Author Response

The author is very pleased to the evaluation by the Reviewer II.